# Skeletal Muscle Mitochondrial Dysfunction and Oxidative Stress in Peripheral Arterial Disease: A Unifying Mechanism and Therapeutic Target

**DOI:** 10.3390/antiox9121304

**Published:** 2020-12-18

**Authors:** Kyoungrae Kim, Erik M. Anderson, Salvatore T. Scali, Terence E. Ryan

**Affiliations:** 1Department of Applied Physiology & Kinesiology, University of Florida, Gainesville, FL 32611, USA; kimk1@ufl.edu; 2Division of Vascular Surgery and Endovascular Therapy, University of Florida, Gainesville, FL 32611, USA; Erik.Anderson@surgery.ufl.edu (E.M.A.); Salvatore.Scali@surgery.ufl.edu (S.T.S.); 3Malcom Randall Veteran Affairs Medical Center, Gainesville, FL 32611, USA; 4Center for Exercise Science, University of Florida, Gainesville, FL 32611, USA

**Keywords:** myopathy, peripheral vascular disease, bioenergetics, ischemia, reactive oxygen species

## Abstract

Peripheral artery disease (PAD) is caused by atherosclerosis in the lower extremities, which leads to a spectrum of life-altering symptomatology, including claudication, ischemic rest pain, and gangrene requiring limb amputation. Current treatments for PAD are focused primarily on re-establishing blood flow to the ischemic tissue, implying that blood flow is the decisive factor that determines whether or not the tissue survives. Unfortunately, failure rates of endovascular and revascularization procedures remain unacceptably high and numerous cell- and gene-based vascular therapies have failed to demonstrate efficacy in clinical trials. The low success of vascular-focused therapies implies that non-vascular tissues, such as skeletal muscle and oxidative stress, may substantially contribute to PAD pathobiology. Clues toward the importance of skeletal muscle in PAD pathobiology stem from clinical observations that muscle function is a strong predictor of mortality. Mitochondrial impairments in muscle have been documented in PAD patients, although its potential role in clinical pathology is incompletely understood. In this review, we discuss the underlying mechanisms causing mitochondrial dysfunction in ischemic skeletal muscle, including causal evidence in rodent studies, and highlight emerging mitochondrial-targeted therapies that have potential to improve PAD outcomes. Particularly, we will analyze literature data on reactive oxygen species production and potential counteracting endogenous and exogenous antioxidants.

## 1. Introduction

Peripheral artery disease (PAD) is a common manifestation of atherosclerosis that affects more than 200 million people worldwide and is the third leading cause of cardiovascular mortality [1]. PAD is caused by atherosclerotic narrowing or occlusion of blood vessels in the lower extremities that leads to a spectrum of life-altering symptomatology, including claudication, ischemic rest pain, and gangrene requiring limb amputation. Morbidity and mortality associated with PAD have increased over the last decade despite an increase in the number of lower extremity vascular procedures during that time. Current treatments for PAD are focused solely on re-establishing blood flow to the limb, implying that blood flow is the definitive factor that determines pathobiology. Unfortunately, failure rates of endovascular and revascularization procedures remain unacceptably high [2,3] and numerous cell- and gene-based vascular therapies have failed to demonstrate efficacy in clinical trials [4,5,6,7]. The low success of vascular-focused therapies implies that other non-vascular tissues must contribute to PAD pathobiology. Although atherosclerotic PAD manifests in the vascular system, there are significant consequences to skeletal muscle that result in impaired muscle health/function and exercise intolerance [8]. Several large clinical studies in PAD patients have demonstrated that muscle function/exercise capacity is the strongest predictor of morbidity/mortality [9,10,11,12,13,14,15]. Previous reports have documented evidence of skeletal muscle myopathies in PAD patients [16,17], although its potential role in clinical pathology is not well understood. In this review, we analyze the evidence that PAD is associated with a skeletal muscle myopathy, including altered mitochondrial function and oxidative stress (Figure 1); discuss the risk factors that contribute to skeletal muscle myopathy; and review emerging therapeutic approaches that aim to mitigate these myopathic symptoms and have potential to improve outcomes for PAD patients.

## 2. Pathogenesis, Diagnosis, and Treatment of PAD

Before embarking upon an analysis of the role of skeletal muscle in PAD pathobiology, a brief discussion of pathogenesis, diagnosis, and treatment of PAD is required. The prevalence of PAD is estimated to be 1–4% in the general population; however, among selected subgroups of elderly patients, rates exceeding 20% have been reported [18]. Due to the frequent association of PAD with cardiovascular risk factors, such as smoking, diabetes, hypertension, dyslipidemia, and increasing age, approximately 200 million patients worldwide have some form of PAD [19]. The pathogenesis of PAD has been linked to several causes, including subendothelial dysfunction, aberrant platelet activity, hyperlipidemia, tobacco exposure, as well as a myriad of other immunologic and inflammatory factors [20,21,22,23,24,25,26]. The protean clinical presentation is underscored by subintimal accumulation of lipid and fibrous materials that form a fixed arterial stenosis and/or occlusion. Classically, lower extremity atherosclerotic occlusive disease involves either the aortoiliac, femoral-popliteal, or tibial vessels, and certain subpopulations, like diabetic patients and/or subjects with renal dysfunction, present with more distal disease [27,28,29]. The clinical manifestation of PAD occurs along a spectrum with some patients being asymptomatic while others describe disabling intermittent claudication (IC). Notably, 5–10% of patients have chronic limb-threatening ischemia (CLTI) and have significantly elevated risk of major limb amputation, adverse cardiac events, and mortality compared to age-matched controls [30,31,32]. The characterization of PAD severity is predominantly dependent upon the constellation of symptoms, with non-invasive studies corroborating the presence of disease. Reproducible exertional lower extremity muscle cramping that is relieved by rest and associated with a fixed arterial stenosis/occlusion is the hallmark description of vasculogenic claudication. In contrast, CLTI may present as rest pain or tissue loss, which classically involves the forefoot. Initial evaluation includes a thorough history and physical with ankle-brachial index (ABI) assessment to confirm the diagnosis.

For patients with claudication, risk of major limb amputation is < 1% per year, but the 5-year major adverse cardiovascular event risk including death approached 30% [33,34]. Therefore, due to the non-limb-threatening nature of this PAD presentation, clinical practice guidelines and peer reviewed evidence strongly advocates that first-line therapy be medical, including smoking cessation, an exercise walking program, hypertension and blood glucose control, daily antiplatelet and statin therapy, as well as a trial of cilostazol [35,36,37]. Subjects that are compliant with initial medical management but have refractory symptoms generally undergo further imaging (e.g., computed tomographic arteriogram, magnetic resonance arteriogram, segmental doppler, and/or pulse volume recording) to delineate the anatomic extent of disease and identify potential treatment options. Most commonly, patients with claudication present with single-level aorto-iliac or superficial femoral artery atherosclerotic occlusive disease and revascularization is usually based upon endovascular strategies if anatomically eligible. However, long-segment occlusive disease is optimally treated with construction of an arterial bypass. Successful outcomes of endovascular or open surgical remediation are determined by improvement in absolute walking distance, patient quality of life, and augmented ABIs.

In contrast to claudication, PAD patients with CLTI require revascularization due to the significant risk of major limb amputation without restoration of limb perfusion [30]. An ankle pressure < 50mmHg, toe pressures < 30mmHg, monophasic Doppler waveforms, low-amplitude calf pulse volume recordings, transcutaneous oxygen pressure < 30mmHg, and/or tissue loss with any objective measurement of occlusive disease all strongly support the clinical diagnosis of CLTI [29]. Limb salvageability is determined using the Society for Vascular Surgery Wound, Ischemia, and foot Infection (WiFi) and Rutherford classification systems [38,39]. Tenets of revascularization include determination of the inflow vessel, the recipient outflow vessel, conduit choice, hemodynamic significance of the lesion, and characterization of the patient’s functional and physiological status. These variables inform the clinician to optimally select the unique arterial reconstruction strategy for each patient. Generally, CLTI patients present with more complex patterns of PAD, so the probability of a durable outcome with endovascular recanalization is less compared to bypass but many providers subscribe to an ‘endo first’ approach [40,41,42]. For subjects undergoing infrainguinal arterial bypass, use of single-segment ipsilateral saphenous vein and clinical stage of the CLTI presentation are the most important determinants of limb salvage [43,44]. Clinical success is highlighted by freedom from major adverse limb events, such as unplanned re-intervention, thrombosis free graft function, and/or major limb amputation.

## 3. Skeletal Muscle Pathology in PAD

The pathogenesis of PAD is undoubtedly vascular in nature with a primary causative factor involving atherosclerosis. However, the ensuing limb muscle pathology appears to be intimately linked to patient outcomes [10,11,13,14,15,45,46,47,48,49]. A large body of literature from clinical human studies reports that altered skeletal muscle phenotype is prevalent in ischemic limbs and that patients with PAD present with reduced exercise capacity and impaired lower extremity skeletal muscle function compared with age-matched healthy counterparts [49,50,51,52,53]. Longitudinal cohort studies determined that the exercise capacity and functional performance are strong predictors of mortality [15,54], implicating skeletal muscle health/function as an important clinical feature of PAD pathobiology.

Limb muscle pathology in PAD manifests as weakness and exercise intolerance. In addition, claudication and/or ischemic rest pain further reduce physical activity levels and negatively impact functional independence and quality of life [55]. The genesis of exercise intolerance in PAD is without doubt multifactorial [55,56]. Blockage of arteries that causes restricted blood flow to lower extremities is amongst the primary factors that negatively affects mobility and functional performance [57]. Further to limb perfusion limitations, walking performance in PAD patients is also related to muscle fiber size [48], and muscle size/strength is predictive of mortality [10,11,13]. Indeed, one of the most notable features of skeletal myopathy related to peripheral arterial insufficiency is lower extremity muscular atrophy. Data from computed tomographic (CT) imaging demonstrated that PAD patients have smaller calf muscle area with accumulated adipose tissue compared to the individuals without PAD [13,57,58]. Further investigation of morphology using histopathologic analysis revealed that a smaller cross-sectional area of both type I and II fibers as well as increased intramuscular collagen deposition [53,57,59]. The underlying mechanisms that explain these changes are complex and still require further examination, but a number of studies in human and animal studies suggest that impaired proteostasis (i.e., altered signaling pathways that regulate protein synthesis and degradation) plays a central role in muscle atrophy [60,61]. A dysfunctional peripheral nervous system, such as identified demyelination or focal denervation on the neuromuscular junction, will also likely stimulate the process of muscle atrophy and contribute to the weakness and functional impairment [62,63]. Skeletal muscle abnormalities also include capillary rarefaction and impaired microcirculation in PAD patients with IC and these findings are more obvious in people suffering from CLTI [59,64,65]. Considering the aspects of limited conduit artery blood flow and impaired endothelium-dependent vasodilation during physical activity in this population, there will probably be an increased reliance on the microcirculation to meet the metabolic demands of the contracting muscle. Therefore, novel strategies targeting skeletal muscle myopathy to reverse muscle atrophy [66,67] that can be used in addition to those targeting capillary rarefaction [68,69] hold great potential to produce greater and more sustainable clinical impact.

Interestingly, following progressive resistance training, PAD patients with intermittent claudication showed delayed onset of claudication time and improved maximum walking time during a graded treadmill test in addition to the increased leg muscle strength [68]. These changes occurred without improving limb hemodynamics (ABI), suggesting the perfusion-independent beneficial impact of resistance training may arise from changes in skeletal muscle. In support of this concept, invasive revascularization in symptomatic PAD patients does not effectively normalize exercise performance [70,71], and acute pharmacological administration of sildenafil, which enhances limb perfusion, did not alter walking capacity [72]. Taken together, these observations indicate that factors other than lower extremity perfusion must play a significant role [8].

## 4. Skeletal Muscle Mitochondrial Function: Evidence from PAD patients

Skeletal muscle has a dynamic range of energy demand that increases dramatically with the onset of contractile activity. Under normal conditions, this energy (ATP) demand is primarily met through the highly efficient process of oxidative phosphorylation within the mitochondria. The mitochondria are responsible for more than 90% of the oxygen consumed in skeletal muscle, which is used to catabolically convert the fuel we consume into an electrochemical gradient that is used to generate ATP and power muscular work. This process involves the coordinated efforts of many enzymes that facilitate oxidation of fuels (i.e., glucose, fats, proteins) and pass electrons into the mitochondrial electron transport system, where the fall in redox potential drives the pumping of protons across the inner mitochondrial membrane from the matrix to the inner membrane space (Figure 2). Electrons from these complex carbons can enter the electron transport system (ETS) via complex I (NADH dehydrogenase), complex II (succinate dehydrogenase), electron-transferring flavoprotein-ubiquinone oxidoreductase (ETFQOR), as well as glycerophosphate and dihydroorotate dehydrogenases. The latter for entry points deliver electrons directly to ubiquinone, bypassing complex I. These events result in generation of an electrochemical gradient of hydrogen ions that powers the ATP synthase to phosphorylate ADP into ATP, which is subsequently transported to the cytosol to power cellular work (i.e., contraction). Because of the large and dynamic range of energy demand in muscle fibers, alterations in mitochondrial function are intimately related to the trajectory of muscle health, function, and disease.

The reduced oxygen delivery to lower limb muscles in PAD undoubtedly compromises these processes and contributes to muscle impairment in these patients. Moreover, pathological changes in mitochondria have been reported in muscle from PAD patients. Using careful analysis of studies on muscle mitochondrial function in PAD, we compiled results from studies that have assessed mitochondrial health using one of two primary methods: (i) assessments of mitochondrial respiration using permeabilized myofibers or isolated mitochondria, and (ii) in vivo assessments with magnetic resonance spectroscopy (Figure 3). We chose these two approaches distinctly because the entire system of mitochondrial energy transduction remains intact for these assessments. This is in stark contrast to studies that have investigated isolated enzyme kinetics in muscle specimens from PAD [47,73], which provide information solely about the activity/capacity of specific single enzymes rather than the complex integration of many enzymes required for energy transformation in mitochondria.

Earlier studies employing phosphorus magnetic resonance spectroscopy for in vivo muscle metabolism measurements reported slow phosphocreatine (PCr) recovery rates following contraction [74,75,76,88,89,90]. This observation is typically consistent with a lower oxidative capacity; however, in the context of PAD, the interpretation is blurred by the presence of impaired oxygen delivery to the measured muscles, which could cause slower PCr recovery. The impact of impaired convective oxygen delivery on the delayed PCr recovery rates in PAD was recently highlighted by the work of Hart et al. [83]. In this study, Hart and colleagues reported lower in vivo maximal ATP synthesis rates in PAD patients (determined via initial PCr resynthesis rates) compared to healthy controls during normal conditions—an observation previously reported by a number of studies [74,75,76,80,87]. Using a reactive hyperemia protocol with supplemental oxygen, they further demonstrated that increasing convective oxygen delivery to limb muscle in PAD patients abolished the difference in maximal ATP synthesis rates between PAD patients and healthy controls. Thus, future studies aiming to employ in vivo assessments of muscle energetics in PAD patients should carefully consider the impact of reduced oxygen delivery when interpreting data.

Using muscle biopsy specimens from PAD patients, previous studies have reported increased levels of mtDNA mutations [91], which have more recently been associated with walking performance in PAD patients [92]. Assessments of mitochondrial respiration in PAD muscle have been performed by a number of groups; however, the findings reported are not uniformly in agreement. For example, Pipinos et al. [78] reported lower ADP-stimulated respiration in muscle of 10 PAD patients when compared to non-PAD controls. Following this initial publication, this same group examined a larger population (*n* = 25 PAD and 16 controls) and observed similar findings of decreased mitochondrial respiration, specifically supported by electron transport system complexes I, III, and IV but not II [79]. Recent studies employing a similar methodology for mitochondrial assessments have reported that age-matched control subjects and mild PAD patients with intermittent claudication have similar muscle mitochondrial respiratory function [82,83,86] or even elevated mitochondrial respiratory capacity normalized to either weight or citrate synthase activity [84]. These contrasting findings are likely explained by the clinical characteristics of patients in the studies. The two studies from Pipinos et al. [78,79] reported that patient clinical characteristics are more akin to patients with CLTI (ABI <0.4 in most patients), a severe manifestation of PAD requiring surgical intervention or limb amputation. Indeed, the impact of PAD severity on muscle mitochondrial function was recently reported in a comparison across non-PAD, intermittent claudicants, and CLTI patients [86]. Collectively, the results of these studies suggest that mild PAD patients likely exhibit near-normal mitochondrial function when compared to appropriate controls, whereas more severe clinical pathology, such as CLTI, is associated with a severe mitochondrial myopathy.

## 5. Skeletal Muscle Oxidative Stress in PAD

The mitochondria have long been identified as a major source of reactive oxygen species (ROS) [93,94,95]. Mitochondrial energy transduction involves an interconnected series of enzymes that transfer electrons from the complex carbon chains we consume (i.e., carbohydrates and fats) to oxygen. This transformation of chemical energy to a redox potential and subsequently a cellular energy charge (ATP/ADP) is analogous to an electrical circuit, wherein increased resistance contributes to elevated electron leak to form ROS. In this regard, ROS are produced by the leak of electrons from donor redox centers to oxygen to form the parent ROS molecule superoxide. More than 10 redox sites have been demonstrated to produce superoxide or hydrogen peroxide within the mitochondrial electron transport system or dehydrogenases within the TCA cycle or β-oxidation (for a detailed description of these sites, please see the following reviews from Martin Brand and colleagues [95,96]). A crucial determinant of the rate of ROS production at each site is the local redox potential (i.e., ratio of the reduced/oxidized electron carrier), and it has been established that the capacity for electron leak varies widely across sites, with complex I_Q_ (quinone binding site of complex I) and complex III_Q_ (quinone binding site of complex III) having the largest capacity. Once superoxide is produced, it is rapidly dismutated to hydrogen peroxide (H_2_O_2_) by superoxide dismutases (SOD) located in the mitochondrial matrix, inner membrane space, and cytosol, as well as the extracellular space. H_2_O_2_ can be enzymatically reduced to water through glutathione and thioredoxin/peroxiredoxin redox circuits that derive reducing power from NADPH [97,98]. Catalase, another antioxidant enzyme that is robustly expressed in skeletal muscle, also degrades H_2_O_2_ to water and oxygen.

In the context of the principles of bioenergetics and redox biology described above, PAD presents several metabolic challenges within mitochondria that contribute to oxidative stress. First, PAD patients have low physical activity levels, which suggests their fuel intake (i.e., amount of food consumed) outpaces their energy requirements. In muscle, this would contribute to increased levels of reduced electron carriers (NADH) compared to oxidized, which has been shown to result in increased mitochondrial ROS production [99]. In this regard, type 2 diabetes is a strong risk factor for PAD and has been shown to worsen myopathic symptoms in ischemia muscle and health outcomes in PAD [28,100,101,102]. In gastrocnemius biopsies, Pipinos et al. [79] reported increased levels of protein carbonyls, lipid hydroperoxides, and 4-hydroxy-2-nonenal (4HNE) in severe PAD patients compared with non-PAD controls. The authors also observed a strong correlation between protein carbonyl levels in muscle and the duration of PAD symptoms. The same group confirmed these findings in a more recent cohort of PAD patients [103], where they further identified correlations with disease severity (both ABI and Fontaine stage), as well as a link between protein carbonyl and 4HNE levels in the myofiber size using histological analyses. Koutakis et al. [104] further reported that carbonylation and 4HNE adducts were elevated in both type I (slow) and type II (fast) myofibers, although the magnitude was larger in type II fibers. While the above work from the Pipinos lab has consistently reported elevated levels of carbonylation and 4HNE in PAD muscle, it is important to note that these outcomes are considered biomarkers of oxidative stress but are not direct measures of ROS. Thus, Hart et al. [82] set out to directly measure ROS in PAD muscle specimens using electron paramagnetic resonance (EPR). In this recent work, the authors observed increased ROS levels by EPR and also confirmed elevated levels of 4HNE and protein carbonyls in PAD muscle specimens. Together, these findings implicate oxidative stress as a key driver of myofiber atrophy, a topic that has been thoroughly reviewed in non-PAD conditions (i.e., cancer cachexia, disuse atrophy, etc.) [105,106].

Another major metabolic challenge facing the ischemic limb muscle centers on the local ischemic environment. The atherosclerotic blockages in arteries supplying the lower limb muscles decrease the capacity for oxygen delivery. During brief bouts of physical activity, such as walking, increased demand for oxygen in the contracting skeletal muscle can outpace the supply, resulting in transient ischemic episodes in muscles distal to the blockage, which can be re-perfused by resting. These repeated bouts of ischemia-reperfusion can become a significant source of pathology (Figure 4). During periods of ischemia, ATP production by oxidative phosphorylation declines as oxygen becomes limiting for the electron transport system. The decline in ATP/ADP (energy charge) occurs during ischemia because of continued ATP utilization to sustain cell functions, such as ion gradients. Moreover, as the electron transport system comes to a halt during ischemia, the proton motive force (i.e., membrane potential) will decline as the proton pumps begin to fail. A decline in the proton motive force can cause the ATP synthase enzyme to operate in reverse and hydrolyze ATP in an attempt to maintain the membrane potential. During reperfusion, mitochondrial ROS have been established as the first damaging event that causes acute damage as well as activating damage-associated molecular pattern molecules (DAMPS) that initiate immune responses, which cause pathology over the ensuing days and weeks following reperfusion. Although the exact cause of mitochondrial ROS during ischemia reperfusion remains complicated, an emerging evidence centers on succinate accumulation during ischemia and reverse electron flow through complex I (NADH dehydrogenase) of the electron transport system during reperfusion [107].

Preclinical rodent studies have reported profound effects of acute ischemia-reperfusion on muscle mitochondrial health. Maximal state 3 (ADP-stimulated) respiration is significantly reduced following a brief period (2–3 h) of ischemia + reperfusion (2 h) in rats [108,109,110,111] and mice [112]. This rapid impairment in mitochondrial respiratory function was associated with elevated mitochondrial ROS production [108,109,112,113]. Although direct evidence of ischemia reperfusion injury in human PAD skeletal muscle is still lacking, there is clear evidence of myofiber injury and metabolic disturbances following tourniquet-induced ischemia-reperfusion in humans undergoing limb surgeries [114,115]. Nonetheless, the repeated bouts of muscular activity and rest that occur throughout normal daily life in PAD patients inevitably causes some level of ischemia-reperfusion events to occur. In fact, near infrared spectroscopy measurements have confirmed that modest walking causes a substantial drop in limb muscle tissue oxygenation [116,117].

The consequences of skeletal muscle oxidative stress in PAD pathobiology and the associated health outcomes in patients are still unclear. Nonetheless, there is a wealth of evidence from the muscle biology literature demonstrating that muscle wasting disorders (i.e., disuse atrophy, cancer cachexia, chronic kidney disease) exhibit redox-sensitive transcriptional networks, including atrogenes and dysregulated autophagy, that play causal roles in muscle atrophy [106,118]. Consistent with observations of elevated biomarkers of oxidative stress in PAD muscle [79,103,104], muscle biopsy specimens from severe PAD patients [119] displayed elevated mRNA levels for atrogenes, autophagy genes, and caspases (Figure 5) compared with non-PAD controls. Thus, a “two-hit” hypothesis emerges, wherein elevated oxidative stress in PAD limb muscles amplifies redox-sensitive atrophy signaling pathways that exacerbate muscle dysfunction and wasting occurring due to low physical activity levels in PAD patients. Clinically, this low muscle mass/function emerges as an increased risk of morbidity and mortality in patients [11,13,15,45,48].

## 6. Preclinical Evidence Supporting a Causal Role of Mitochondria in the Pathogenesis of Limb Ischemia

Preclinical studies have recently identified divergent limb responses to surgically induced limb ischemia in inbred strains of mice [120,121,122,123]. Specifically, it has been established that inbred strains, such as the C57BL6J, exhibit a rapid and near linear recovery of limb perfusion with minimal tissue and muscle necrosis. In contrast, BALB/c mice experience poor perfusion recovery, substantial muscle injury, and necrosis in response to the same surgical intervention [120,121,124,125]. Before discussing evidence supporting a causal role of muscle mitochondria in ischemia myopathy, it is important to note that rodent models of limb ischemia (femoral artery ligation) result in an abrupt and severe decrease in limb perfusion, which is not directly representative of the human pathology. Nonetheless, the utility of these models stems from the consistent outcomes from the surgery performed with rigorous standards and the ability to manipulate genes and deliver experimental pharmaceuticals that are not possible in human patients. This facilitates testing of causal roles in the pathogenesis of limb ischemia.

Interestingly, Schmidt et al. [122] reported that BALB/c, but not C57BL6J, mice experience significant decreases in mitochondrial respiratory capacity and calcium retention capacity even with a mild limb ischemia model induced by ameroid constrictors that produced identical levels of tissue oxygen saturation in both strains. Further, catastrophic mitochondrial impairment occurs within the first 24 h of ischemia in BALB/c mice and fragile mitochondrial respiratory function remains present for at least 56 days following hindlimb ischemia despite significant increases in limb perfusion [126]. Together, these findings implicate a divergent response in mitochondrial energetics in mouse strains with high (BALB/c) and low (C57BL6J) susceptibility to ischemic myopathy but fall short of establishing causal evidence. To address this, we recently asked whether imparting mitochondrial dysfunction in C57BL6J would exacerbate ischemic outcomes [126]. To accomplish this, the polymerase gamma mutant mouse, which accumulates mtDNA mutations, was bred on the C57BL6J genetic background and aged to 12 months prior to femoral artery ligation. Although the Polg^D257A^ mutant mice exhibited impaired mitochondrial energetics, they were shockingly protected from ischemic muscle injury as a result of an increased capacity for non-aerobic metabolism stemming in part from 6-phosphofructo-2-kinase/fructose-2,6-bisphosphatase 3 (PFKFB3) expression, which is a known allosteric activator of glycolysis [126]. These findings suggest that fragile mitochondrial health in muscle, if compensated for by a robust reprogramming of non-aerobic metabolic pathways, results in non-pathogenic outcomes. Nonetheless, this study found that muscle from severe PAD patients and BALB/c mice (akin to CLI patient) fail to accomplish this metabolic programming and suffer from severe ischemic myopathy, highlighting the crucial role of metabolic homeostasis in the ischemic limb.

Preclinical studies of targeting mitochondria provide support for a causal role in ischemic limb outcomes. For example, expression of a mitochondrial-targeted catalase, which degrades hydrogen peroxide to water and decreases oxidative stress, was found to substantial improve limb muscle function and decrease ischemic injury in mice with type 2 diabetes [102]. Similarly, treatment with mitoTEMPO, a mitochondrial-targeted superoxide scavenger, in aged mice was found to improve muscle mitochondrial respiration, decrease mtDNA damage, and rescue limb perfusion recovery [127]. Similarly, Lejay et al. [128] reported that treatment with *N*-acetylcysteine, a global antioxidant, also improved mitochondrial respiration and limb muscle outcomes in mice subjected to chronic limb ischemia. More recently, exogenous mitochondrial transplantation was also reported to decrease infarct size and improve gait function in mice subjected to acute (24 h) limb ischemia [129]. It is worth noting that the molecular mechanisms underlying the reported benefits of exogenous mitochondrial transplantation are unresolved and questions have been raised as to whether these transplanted mitochondria are able to maintain bioenergetic function in the high calcium extracellular environment [130]. Using global mitochondrial aldehyde dehydrogenase 2 (ALDH2) knockout mice, Liu at el. [131] reported poor perfusion recovery in ALDH2-KO mice, which was associated with impaired angiogenic signaling, further substantiating a role of mitochondria in ischemic outcomes. Taken together, these studies provide compelling evidence that mitochondria within the ischemic limb play a pivotal role in the pathobiology of PAD.

## 7. Risk Factors and Comorbidities Converge on Muscle Mitochondria

Complicating the etiology of PAD, patients typically present with one or more comorbid conditions or risk factors that accelerate disease evolution and are associated with poorer health outcomes. Among these, tobacco smoking, diabetes and obesity, and chronic kidney disease (CKD) all independently have negative impacts on skeletal muscle mitochondrial function [99,132,133,134,135,136,137]. Tobacco smoking is the most lethal risk factor for PAD and also the most important risk factor for chronic obstructive pulmonary disease (COPD). Patients with COPD present with a strikingly similar pathophysiology to PAD patients, including exercise intolerance, decrease muscle strength and endurance, decreased mitochondrial respiratory function, and elevated muscle ROS levels (reviewed in [138]). The etiology of muscle dysfunction in COPD patients (similar to PAD patients) is multifactorial with multiple factors, including hypoxemia, physical inactivity, and inflammation. Likewise, type 2 diabetes and obesity are strong risk factors for PAD and are also independently associated with abnormal skeletal muscle mitochondrial function (principally elevated ROS) [132,139,140,141]. CKD is another major risk factor for PAD, although the pathological basis is far less understood compared to smoking and diabetes. CKD accelerates atherosclerosis [142] and is highly prevalent in PAD patients [143]. In a similar fashion, CKD alone has also been demonstrated to negatively impact skeletal muscle mitochondrial function [134,136,144,145]. Coincidently, each of these risk factors has been strongly linked to higher mortality and amputation risk in PAD patients, as well as an increased rate of failure of endovascular and revascularization procedures in PAD [28,146,147,148,149,150,151,152]. These observations raise the intriguing hypothesis that skeletal muscle mitochondria may represent a critical point of convergence of pathophysiology from PAD and the associated co-morbid conditions.

## 8. Therapeutic Targeting of Mitochondria: The Future Is Bright for PAD

There is growing interest in the development of pharmacological approaches to modulate mitochondrial function as a means of treatment of disease (thoroughly reviewed in [153,154,155]). Current and emerging approaches focus on increasing ATP synthesis, reducing or scavenging ROS, reducing permeability transition events, and inducing mitochondrial biogenesis. Several naturally occurring substances, including co-enzyme Q, idebenone, and dimethylglycine, have been demonstrated to improve mitochondrial ATP synthesis in patients with various mitochondrial disorders [156,157,158,159]. Additional compounds have been generated to reduce ROS production or directly scavenge ROS to reduce the burden of oxidative stress associated with mitochondrial pathologies [160,161,162,163,164]. Given that oxidative stress and impairment of oxidative phosphorylation are clearly present in the human PAD skeletal muscle, mitochondrial-targeted therapies may be an untapped area of therapeutic investigation. Recent preclinical evidence has demonstrated that treatment with antioxidant therapies show promise in both chronic ischemia and acute ischemia-reperfusion models (Table 1). For example, Lejay et al. reported that *N*-acetylcysteine, a global antioxidant and precursor to glutathione, improved mitochondrial respiratory function and reduced oxidative stress in mice subjected to chronic severe ischemia in both normal Swiss mice [128] and apolipoprotein E-deficient mice [165]. Another study employed a transgenic mouse overexpressing a mitochondrial-targeted catalase (scavenger of hydrogen peroxide), which was shown to decrease ischemia muscle injury and improve mitochondrial function in diet-induced diabetic mice [102]. Similarly, treatment with elampretide, a cardiolipin-targeting tetrapeptide [166], was also shown to improve ischemic myopathy in mice with critical limb ischemia [167]. In models of acute ischemia reperfusion, treatment with cyclosporine A, an immunosuppressant that inhibits calcineurin and the mitochondrial permeability transition pore [108,111], and MitoSNO (a mitochondrial S-nitrosylation enhancer) [168] were also shown to be efficacious. Although these preclinical studies raise enthusiasm for antioxidant therapies, it is important to consider the potential of these treatments to impact exercise adaptations in muscle [169].

To date, the translation of these promising preclinical results to the clinic has been slow (Table 1). In fact, there have been only a few clinical studies involving mitochondrial-targeted therapies in PAD patients. Following promising results in several smaller clinical studies [170,171,172], a multicenter trial involving 485 PAD patients demonstrated that treatment with L-carnitine improved walking distance in more severe claudicating patients but had no effect on patients with more mild disease [173]. These results were confirmed by a second trial that also reported increased walking distance and speed in claudicating PAD patients [174]. More recently, McDermott and colleagues [175] reported that daily consumption of a cocoa flavanol beverage (containing 75mg of epicatechin—a plant-based antioxidant) for 6 months improved 6-min walk performance and increased cytochrome c oxidase (complex IV of the ETS) activity in calf muscle homogenates. Another recent study by Park et al. [176] examined the impact of mitoQ (a mitochondrial-targeted antioxidant) supplementation in PAD patients (Fontaine stage II and III). Using a cross-over placebo-controlled trial, the authors observed that mitoQ improved endothelial function (flow-mediated dilation) and walking performance in these mild PAD patients.

## 9. Concluding Remarks and Perspectives

Worldwide, the population of PAD patients continues to grow, and this trend is expected to continue in the coming decades. Surgical interventions for PAD have increased substantially in numbers and have improved substantially from a technical perspective over the past two decades. However, the mortality and morbidity rates in PAD patients has not decreased proportionately. Coupled with the lack of efficacious drug therapies and cell/gene-based angiogenic therapies for PAD, there is a dire need to develop more effective treatments for this serious disease. Skeletal muscle dysfunction, including sarcopenia/muscle wasting, mitochondrial dysfunction, and oxidative stress, are consistently reported across the clinical PAD literature and are the strongest predictors of mortality and health outcomes. With the continued growth and refinement of mitochondrial-targeting therapies, further investigation in PAD patients holds tremendous promise to improve clinical outcomes and provide hope to patients with limited available treatment options.

## Figures and Tables

**Figure 1 antioxidants-09-01304-f001:**
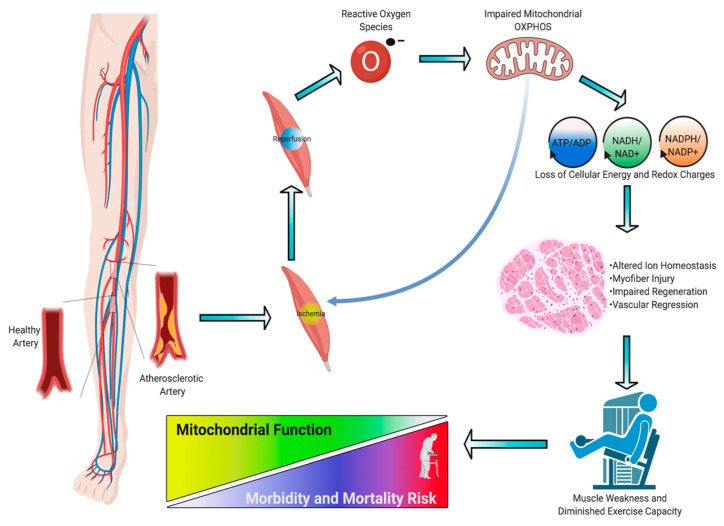
Pathogenesis of ischemic myopathy in peripheral artery disease (PAD). A graphical depiction of the pathogenesis of ischemic myopathy in PAD. This figure was created with Biorender.com.

**Figure 2 antioxidants-09-01304-f002:**
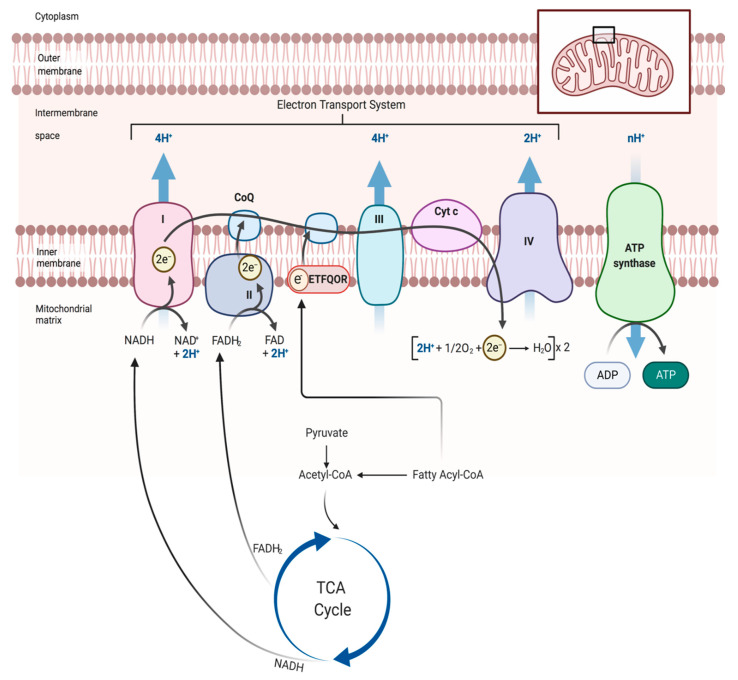
Overview of mitochondrial energetics. A graphical overview of the mitochondrial energy transduction pathway. Electrons are transferred from complex carbon changes (sugars, proteins, fats) to carrier molecules (NAD^+^ and FAD^+^), which feed the electrons in the electron transport system (ETS) located in the inner mitochondrial membrane. The ETS transfers electrons across several protein complexes down a redox potential to oxygen. Consequently, the fall in redox potential powers the proton pumping capabilities of complexes I, III, and IV, which establish the proton motive force (mitochondrial membrane potential). The proton motive force is then used by the ATP synthase to drive phosphorylation of ADP to form ATP and establish the cellular energy charge (ATP/ADP), which powers all cellular work. Abbreviations: CoQ = ubiquinone/co-enzyme Q; TCA = tricarboxylic acid; ETFQOR = electron transferring flavoprotein-ubiquinone oxidoreducatase; Cyt C = cytochrome c. This figure was created with Biorender.com.

**Figure 3 antioxidants-09-01304-f003:**
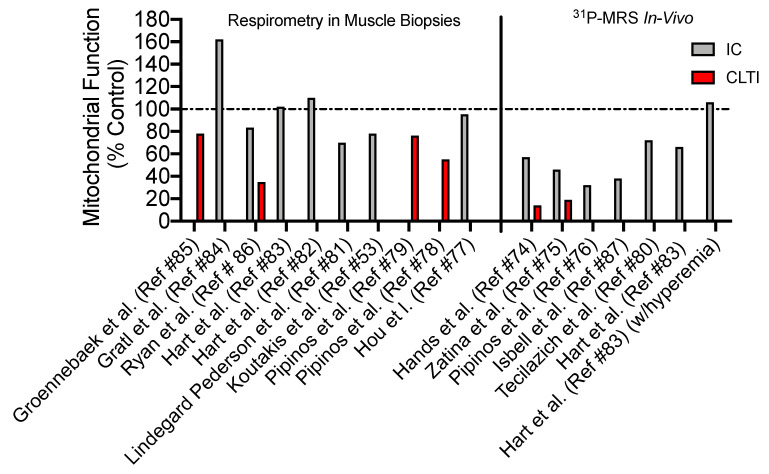
Summary of muscle mitochondrial assessments in PAD patients. A summary of studies assessing mitochondrial function via respirometry in muscle biopsy specimens of in vivo using ^31^P magnetic resonance spectroscopy (MRS). Data are presented as a percentage of the control group and patients were classified as intermittent claudicants (IC—mild PAD) or chronic limb-threatening ischemia (CLTI—severe PAD) based on the clinical characteristics provided by the authors of each paper. Please refer to the following references for these original studies: [53,74,75,76,77,78,79,80,81,82,83,84,85,86,87].

**Figure 4 antioxidants-09-01304-f004:**
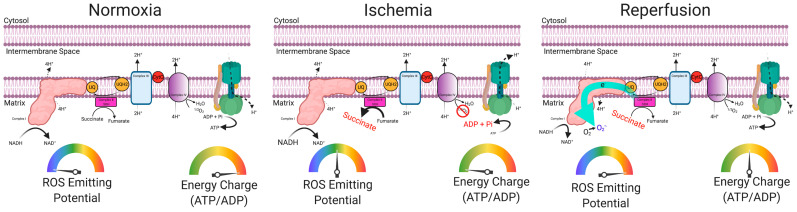
Ischemia-reperfusion injury as a source of oxidative stress in PAD skeletal muscle. A graphical depiction of the mitochondrial electron transport system (ETS) during normoxic conditions, ischemia conditions, and reperfusion conditions. Under normoxia, the functions to transfer potential energy from electrons on NADH/succinate to a proton motive force, which drives ATP synthesis to establish a high energy charge. During periods of ischemia, the low oxygen levels halt electron flow in the ETS due to the lack of oxygen availability, and subsequently, succinate accumulates and the cellular energy charge declines. With reperfusion of ischemic muscle, high levels of succinate oxidation result in reverse electron flow through complex I to drive high levels of ROS production. This figure was created with Biorender.com.

**Figure 5 antioxidants-09-01304-f005:**
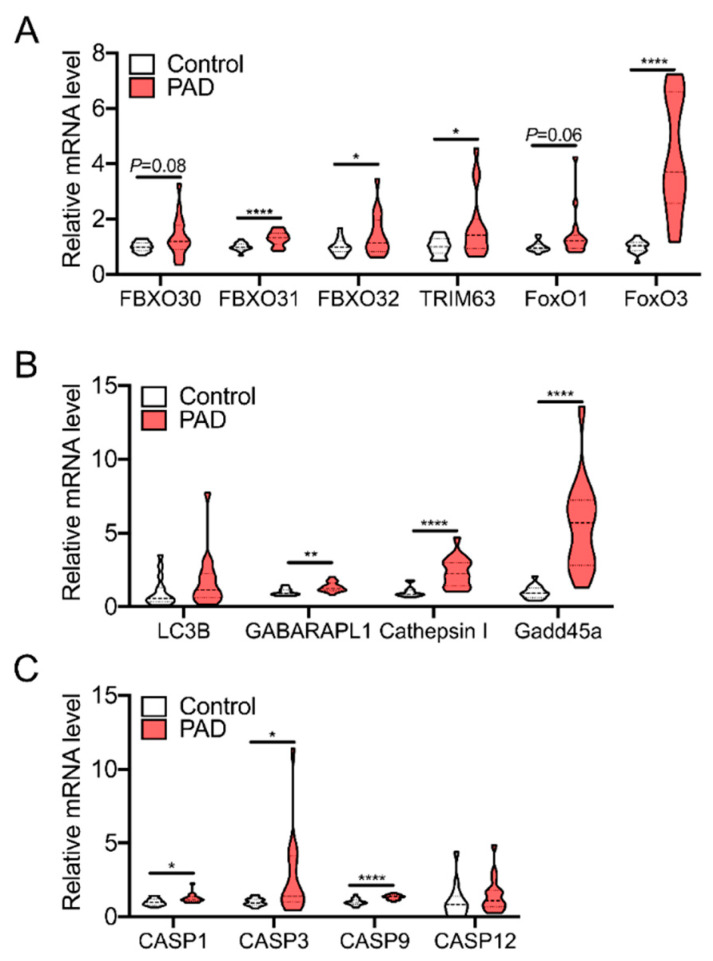
Redox-sensitive transcriptional signaling for muscle atrophy in PAD. Muscle biopsy specimens were collected from non-PAD control (Age = 72.8 ± 7.8 years, ABI = 1.2 ± 0.2) and severe PAD patients undergoing limb amputation (Age = 69.5 ± 6.2 years, ABI = 0.5 ± 0.3) following obtainment of informed consent (study approved by the IRB at the University of Florida). Other clinical characteristics of patients are described in [119]. RNA was isolated using a Qiagen RNeasy minikit and reverse transcribed into cDNA using Superscript IV (Thermofisher). Quantitative RT-PCR was performed using pre-validated Taqman probes (Thermofisher) to assess the levels of redox-sensitive atrogenes (**A**), autophagy genes (**B**), and caspases (**C**) known to be involved in muscle atrophy. Violin plots are shown with dashed lines representing the median and dotted lines the quartiles. *n* = 15/group. * *p* < 0.05, ** *p* < 0.01, **** *p* < 0.001 using two-tailed, unpaired Student’s *t*-test. Full Gene names are as follows: FBXO30 = F-box only protein 30; FBXO31 = F-box only protein 31; FBXO32 = F-box only protein 32; TRIM63 = tripartite motif containing 63 or muscle-specific RING finger; FoxO1 = forkhead box O1; FoxO3 = forkhead box O3; LC3B = microtubule associated protein 1 light chain 3 beta; GABARAPAL1 = GABA type A receptor associated protein like 2; Gadd45a = growth arrest and DNA damage inducible alpha; CASP1 = caspase 1; CASP3 = caspase 3; CASP9 = caspase 9; CASP12 = caspase 12.

**Table 1 antioxidants-09-01304-t001:** Summary of mitochondrial-targeted therapies in preclinical and clinical PAD. A brief summary of selected studies testing the efficacy of mitochondrial-targeted therapies using preclinical rodent models of PAD and human clinical trials in PAD patients.

Preclinical Studies in Rodents			
Reference	Species and PAD Model	Treatment	Main Results
Lejay et al. [128]	Swiss mice—femoral artery ligation 40 days post-surgery	*N*-acetylcysteine	-decrease tissue damage -improved mitochondrial respiration and calcium retention -decreased ROS levels
Lejay et al. [165]	Apolipoprotein E deficient mice—femoral artery ligation 40 days post-surgery	*N*-acetylcysteine	-improved mitochondrial respiration and calcium retention -decreased ROS production
Miura et al. [127]	Mice—femoral artery ligation 21 days post-surgery	MitoTEMPO	-improved limb perfusion recovery -decreased ROS production -decreased mtDNA damage
Pottecher et al. [111]	Wistar rats (young)—acute ischemia (3 h) and reperfusion (2 h)	Cyclosporin A	-improved mitochondrial respiration -decreased ROS production
Pottecher et al. [108]	Wistar rats (old)—acute ischemia (3 h) and reperfusion (2 h)	Cyclosporin A	-no rescue of mitochondrial respiration -increased ROS production
Ryan et al. [102]	C57BL6J mice—femoral artery ligation 7 days post-surgery	Transgenic overexpression of mitochondrial-targeted catalase	-reduced ischemic muscle injury and limb necrosis -improved ischemic muscle contractile function -improved mitochondrial respiration -decreased ROS levels
Ryan et al. [167]	BALB/c mice—femoral artery ligation 7 days post-surgery	Elampretide	-decreased limb necrosis -Improved mitochondrial respiration -increase limb perfusion recovery and capillary density
Wilson et al. [168]	Mice—acute ischemia (1 h) and 7–14 days post-ischemia	MitoSNO	-increased muscle contractile function -decreased muscle denervation
Clinical Studies in Human PAD Patients			
Brevetti et al. [170,171,172]	IC patients (*n* = 8) IC patients (*n* = 10) IC patients (*n* = 30)	l-carnitine	-did not change ABI -improved walking distance
Brevetti et al. [173]	IC Patients (*n* = 485)	Propionyl-l-carnitine	-improved walking distance in severe claudicants but not mild
McDermott et al. [175]	IC Patients (*n* = 44)	Epicatechin	-improved 6-min walk performance-increase cytochrome c oxidase activity -improved muscle perfusion and capillary density
Park et al. [176]	IC Patients (*n* = 22)	mitoQ	-improved endothelial function -improved walking distance and time -delayed onset of claudication

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
