# Peer review of "Skeletal Muscle Mitochondrial Dysfunction and Oxidative Stress in Peripheral Arterial Disease: A Unifying Mechanism and Therapeutic Target"

_antioxidants, 2020, doi:10.3390/antiox9121304_

Round 1

Reviewer 1 Report

The review article antioxidants-1038434 entitled“ Skeletal muscle mitochondrial dysfunction and  oxidative stress in peripheral arterial disease: A unifying mechanism and therapeutic target” by Kyoungrae Kim, and co-workers is definitely a comprehensive up-to date and in general well-written review focusing on a very relevant topic of clinical interest. In particular the authors highlighted and discussed the emerging role of skeletal muscle mitochondrial dysfunction and elevated oxidative stress in peripheral arterial disease, that may open new and promising therapeutic avenues for this pathology management. Nevertheless, there are some aspects to revise and improve.

  1. Overall the review in well-written and organized. However it seems to me that that each paragraph was written “independently” as many concepts are repeated and the references related to the same issues are too much. I suggest to revise the entire manuscript and to remove some references. The  references 105 and 138 as well as 112 and 117  are the same

105 Anderson, E.J., et al., Mitochondrial H2O2 emission and cellular redox state link excess fat intake to insulin 701 resistance in both rodents and humans. J Clin Invest, 2009. 119(3): p. 573-81.

138 Anderson, E.J., et al., Mitochondrial H2O2 emission and cellular redox state link excess fat intake to insulin 772 resistance in both rodents and humans. The Journal of clinical investigation, 2009. 119(3): p. 573-81.

112 Pottecher, J., et al., Skeletal muscle ischemia-reperfusion injury and cyclosporine A in the aging rat. Fundam 713 Clin Pharmacol, 2016. 30(3): p. 216-25.

117 Pottecher, J., et al., Skeletal muscle ischemia-reperfusion injury and cyclosporine A in the aging rat. 724 Fundamental & Clinical Pharmacology, 2016. 30(3): p. 216-225.

  1. Page 2 lines 49-50. Please change the reference 18. Accumulating evidence…reference year 1993
  2. Abbreviations/acronyms should be defined in parentheses the first time they appear in main text and used consistently thereafter. Page 4 line146: IC is not defined. I suggest to define IC in the paragraph 2 e.g. page 3 lines 75-77.
  3. The Figure 1 is not cited in the main text. However this figure which is more a graphical abstract, should me moved forward. Moreover I suggest to add in the figure muscle atrophy e to detail the legend.
  4. Paragraph 4 and Figure 3. There is not a full match between the references cited in this paragraph and those reported in the figure. Please check and correct it. Moreover I suggest to indicate the reference number in the figure and not just name year and journal. Pederson et al. 2017 is not included in the reference list.
  5. Paragraph 5 page 7 lines 242-258. As the authors provided a figure on the well-known mitochondrial energetics they should provide a figure also for ROS production or include this information in Figure 2.
  6. Figure 4 must be enlarged so that it is readable
  7. Figure 5. It is not clear to this reviewer whether data reported in the Figure 5 are your new unpublished data or published ones. If published, as I suppose,  please indicate in the figure legend the references.
  8. Reference list. References are not described according MDPI journal style. See the MDPI Reference List and Citations style Guide for more detailed information.
  9. Through the text there are some typos- spaces before square brackets, that need to be corrected

Author Response

The review article antioxidants-1038434 entitled“ Skeletal muscle mitochondrial dysfunction and  oxidative stress in peripheral arterial disease: A unifying mechanism and therapeutic target” by Kyoungrae Kim, and co-workers is definitely a comprehensive up-to date and in general well-written review focusing on a very relevant topic of clinical interest. In particular the authors highlighted and discussed the emerging role of skeletal muscle mitochondrial dysfunction and elevated oxidative stress in peripheral arterial disease, that may open new and promising therapeutic avenues for this pathology management. Nevertheless, there are some aspects to revise and improve.

  1. Overall the review in well-written and organized. However it seems to me that that each paragraph was written “independently” as many concepts are repeated and the references related to the same issues are too much. I suggest to revise the entire manuscript and to remove some references. The  references 105 and 138 as well as 112 and 117  are the same

105 Anderson, E.J., et al., Mitochondrial H2O2 emission and cellular redox state link excess fat intake to insulin 701 resistance in both rodents and humans. J Clin Invest, 2009. 119(3): p. 573-81.

138 Anderson, E.J., et al., Mitochondrial H2O2 emission and cellular redox state link excess fat intake to insulin 772 resistance in both rodents and humans. The Journal of clinical investigation, 2009. 119(3): p. 573-81.

112 Pottecher, J., et al., Skeletal muscle ischemia-reperfusion injury and cyclosporine A in the aging rat. Fundam 713 Clin Pharmacol, 2016. 30(3): p. 216-25.

117 Pottecher, J., et al., Skeletal muscle ischemia-reperfusion injury and cyclosporine A in the aging rat. 724 Fundamental & Clinical Pharmacology, 2016. 30(3): p. 216-225.

We would like to thank Reviewer #1 for the helpful and positive feedback on this review manuscript.  We apologize for the mistake on the duplicate references – this seems to have been a communication error across EndNote libraries.  We have corrected this in the revised manuscript and removed portions of some sections that we agreed were somewhat repeated in other sections.

  1. Page 2 lines 49-50. Please change the reference 18. Accumulating evidence…reference year 1993

We have revised this to correct the tense/phrasing of this sentence as recommended.

  1. Abbreviations/acronyms should be defined in parentheses the first time they appear in main text and used consistently thereafter. Page 4 line146: IC is not defined. I suggest to define IC in the paragraph 2 e.g. page 3 lines 75-77.

We apologize for this oversight. We have defined IC on Page 3 as recommended.

  1. The Figure 1 is not cited in the main text. However this figure which is more a graphical abstract, should me moved forward. Moreover I suggest to add in the figure muscle atrophy e to detail the legend.

We now callout Figure 1 on Page 2 (line 56).  We have also added substantial detail in the legend of Figure 5 and agree these details are crucial to understand the context of those data.

  1. Paragraph 4 and Figure 3. There is not a full match between the references cited in this paragraph and those reported in the figure. Please check and correct it. Moreover I suggest to indicate the reference number in the figure and not just name year and journal. Pederson et al. 2017 is not included in the reference list.

We have corrected the mistakes with all references and now include the reference citation within the legend of the Figure 3 as well.

  1. Paragraph 5 page 7 lines 242-258. As the authors provided a figure on the well-known mitochondrial energetics they should provide a figure also for ROS production or include this information in Figure 2.

We appreciate the reviewer’s suggestion about adding a figure related to ROS production.  To this end, we chose to focus ROS production efforts in the mechanisms of repeated ischemia reperfusion which is presented in Figure 4.  We agree that the many possible sites of ROS production which include several quinone and Flavin sites within the electron transport system and matrix dehydrogenases.  These sites have been defined and reviewed extensive in a long list of publications from Martin Brand’s Lab (Buck Institute).  We chose to not include a detailed figure listing these site and have referred readers to this papers in the current work.  Our reasoning behind this stems from the lack of information in PAD muscle about which ROS producing sites are altered – all current PAD data simply indicate markers of oxidative stress, but individual site have not been analyzed.  We have expanded some discussion related to this in section #5.

  1. Figure 4 must be enlarged so that it is readable

We have increased the size of Figure 4 as suggested.

  1. Figure 5. It is not clear to this reviewer whether data reported in the Figure 5 are your new unpublished data or published ones. If published, as I suppose, please indicate in the figure legend the references.

We apologize for the confusion regarding data in Figure 5. Major clinical characteristics of the patients was reported in a previous publication (Berru et al. Scientific Reports 2019), but these atrogene RT-PCR data were not reported in that paper. We have revised the manuscript and figure legend to include sufficient detail for these data and patient samples.

  1. Reference list. References are not described according MDPI journal style. See the MDPI Reference List and Citations style Guide for more detailed information.

We thank the reviewer for bringing this to our attention.  We have revised the references to meet the guidelines using the MDPI EndNote style output.

  1. Through the text there are some typos- spaces before square brackets, that need to be corrected

We have corrected these errors and thank the reviewer to identifying these.

Reviewer 2 Report

Kim et al. provide an interesting and comprehensive review about the role of mitochondrial dysfunction in ischemic muscles in the context of peripheral artery disease. Traditionally, reduced blood flow has been considered the decisive, if not the exclusive, factor leading to pathology associated with the peripheral artery disease (PAD). Kim et al. make a convincing argument that other factors should also be considered in order to understand limited success of current therapeutic approaches and to develop new ways to treat PAD. Further, the text is supported by nice figures, which make the reading easier.

Minor comments:

  • Figure 2: It is appreciated that Figure 2 includes electron entry into the electron transport chain at the level of ETF, especially because this point of entry is often omitted. However, it would be better for a general reader to at least mention in a sentence or two what ETFQOR is (abbreviation is not explained).
  • Figure 3: the x-axis should be checked: in some cases journals are written, while in others are not. In one case hyperaemia is indicated, but other references do not include experimental conditions. I would suggest to use the same approach for all references in this figure.
  • “H2O2 can be enzymatically buffered through glutathione and thioredoxin/peroxiredoxin redox circuits that derive reducing power from NADPH.” For a general reader this sentence would need a little more explanation. What does it mean enzymatically buffered? Perhaps the catalase reaction should also be mentioned.
  • “During periods of ischemia, ATP production by oxidative phosphorylation declines as oxygen becomes limiting for the electron transport system. The decline in ATP/ADP (energy charge) occurs during ischemia despite continued ATP utilization to sustain cell functions, such as ion gradients.” The decline in ATP/ADP ratio probably happens because of continued ATP utilization and not despite it? Reduction in the ATP utilization would alleviate the drop in the ATP/ADP ratio.
  • Figure 5: in several cases only abbreviations (without explanation) of gene names are used in graphs 5A-C. It would be appreciated, if a full name of the gene could be included in the figure legend. This would make the text more reader-friendly.
  • PFKFB3: if PFKFB3 is discussed its name should be written in full (only the abbreviation is used) and its function should at least be mentioned.
  • COPD: it should be mentioned that factors leading to muscle dysfunction in COPD are complex and include increased cytokine levels due to chronic inflammation, hypoxaemia, physical inactivity due to lung dysfunction etc.
  • Antioxidants are discussed in the context of PAD therapy. However, antioxidants may also have unwanted effects. E.g. they may decrease (beneficial) responses to exercise. This aspect should be at least mentioned.
  • Cyclosporine A: for a general reader it should also be mentioned that cyclosporine A is an immunosuppressant that inhibits calcineurin. (In order to avoid a potential misunderstanding that it is a specific inhibitor of the mitochondrial permeability transition pore.)
  • The findings of therapeutic compound trials would be easier to follow if they were arranged in a table (subchapter 8).
  • Since this review deals with PAD and skeletal muscle ischemia, I would suggest to the authors to include at least a few sentences or a paragraph to the hypoxia-inducible factor (HIF). HIF has an important role in regulating mitochondria and angiogenesis. The review would gain if HIF is put into the context.

Author Response

Kim et al. provide an interesting and comprehensive review about the role of mitochondrial dysfunction in ischemic muscles in the context of peripheral artery disease. Traditionally, reduced blood flow has been considered the decisive, if not the exclusive, factor leading to pathology associated with the peripheral artery disease (PAD). Kim et al. make a convincing argument that other factors should also be considered in order to understand limited success of current therapeutic approaches and to develop new ways to treat PAD. Further, the text is supported by nice figures, which make the reading easier.

Minor comments:

  • Figure 2: It is appreciated that Figure 2 includes electron entry into the electron transport chain at the level of ETF, especially because this point of entry is often omitted. However, it would be better for a general reader to at least mention in a sentence or two what ETFQOR is (abbreviation is not explained).

We thank the reviewer for this positive and helpful feedback.  We agree that including a sentence describing this is important and we have added this on Page 4 Lines 182-186.  Additionally, we have expanded the figure legend to include abbreviations for readers.

  • Figure 3: the x-axis should be checked: in some cases journals are written, while in others are not. In one case hyperaemia is indicated, but other references do not include experimental conditions. I would suggest to use the same approach for all references in this figure.

We agree with the reviewer that our x-axis labeling could be more standardized.  In the revised figure 3, we include the author, year, and reference number.  In addition, one study we still include the term hyperemia because the measurement was made under both normal and hyperemic conditions as the authors tried to assess potential blood flow limitations within these 31P-MRS measurements made in vivo.

  • “H2O2can be enzymatically buffered through glutathione and thioredoxin/peroxiredoxin redox circuits that derive reducing power from NADPH.” For a general reader this sentence would need a little more explanation. What does it mean enzymatically buffered? Perhaps the catalase reaction should also be mentioned.

We thank the reviewer for this suggestion. We have included a more specific description of these buffering systems that reduce H2O2 to water. We agree that inclusion of catalase is important and have added this to the paragraph as well (Page 7 Line 275).

  • “During periods of ischemia, ATP production by oxidative phosphorylation declines as oxygen becomes limiting for the electron transport system. The decline in ATP/ADP (energy charge) occurs during ischemia despite continued ATP utilization to sustain cell functions, such as ion gradients.” The decline in ATP/ADP ratio probably happens because of continued ATP utilization and not despite it? Reduction in the ATP utilization would alleviate the drop in the ATP/ADP ratio.

We thank the reviewer for catching this grammatical error. The reviewer is correct and we have changed the wording from “despite” to “because of”.

  • Figure 5: in several cases only abbreviations (without explanation) of gene names are used in graphs 5A-C. It would be appreciated, if a full name of the gene could be included in the figure legend. This would make the text more reader-friendly.

Full gene names are now listed in the figure legend as suggested.

  • PFKFB3: if PFKFB3 is discussed its name should be written in full (only the abbreviation is used) and its function should at least be mentioned.

We apologize for this oversite.  We include the full name and description of PFKFB3’s function in the revised paper at Page 10 Lines 405-406.

  • COPD: it should be mentioned that factors leading to muscle dysfunction in COPD are complex and include increased cytokine levels due to chronic inflammation, hypoxaemia, physical inactivity due to lung dysfunction etc.

These are excellent points and well taken by the authors.  We agree the muscle dysfunction in COPD is multifactorial and the reviewers brings attention to numerous factors contributing.  We have revised the manuscript to acknowledge these sources as well (page 11 Line 442-443).

  • Antioxidants are discussed in the context of PAD therapy. However, antioxidants may also have unwanted effects. E.g. they may decrease (beneficial) responses to exercise. This aspect should be at least mentioned.

This is another great point and we agree it is important to highlight this. We have added a sentence and reference related to this potential confounding effect of antioxidants for PAD patients undergoing exercise therapies which are clearly important for mild PAD patients.

  • Cyclosporine A: for a general reader it should also be mentioned that cyclosporine A is an immunosuppressant that inhibits calcineurin. (In order to avoid a potential misunderstanding that it is a specific inhibitor of the mitochondrial permeability transition pore.)

This is an excellent point and we agree CsA cannot be considered to act solely on the permeability transition pore.  We have revised this sentence to include a statement regarding calcineurin.

  • The findings of therapeutic compound trials would be easier to follow if they were arranged in a table (subchapter 8).

We agree with the reviewer and have added the table below to subchapter 8.

Preclinical Studies in Rodents

Reference

Species and PAD Model

Treatment

Main Results

Lejay et al. [Ref #128]

Swiss mice – femoral artery ligation 40 days post-surgery

N-acetylcysteine

-decrease tissue damage

-improved mitochondrial respiration and calcium retention

-decreased ROS levels

Lejay et al. [Ref #165]

Apolipoprotein E deficient mice – femoral artery ligation 40 days post-surgery

N-acetylcysteine

-improved mitochondrial respiration and calcium retention

-decreased ROS production

Miura et al. [Ref #127]

Mice - femoral artery ligation 21 days post-surgery

MitoTEMPO

-improved limb perfusion recovery

-decreased ROS production

-decreased mtDNA damage

Pottecher et al. [Refs #111]

Wistar rats (young)– acute ischemia (3hr) and reperfusion (2hr)

Cyclosporin A

-improved mitochondrial respiration

-decreased ROS production

Pottecher et al. [Refs #108]

Wistar rats (old)– acute ischemia (3hr) and reperfusion (2hr)

Cyclosporin A

-no rescue of mitochondrial respiration

-increased ROS production

Ryan et al. [Ref #102]

C57BL6J mice - femoral artery ligation 7 days post-surgery

Transgenic overexpression of mitochondrial-targeted catalase

-reduced ischemic muscle injury and limb necrosis

-improved ischemic muscle contractile function

-improved mitochondrial respiration

-decreased ROS levels

Ryan et al. [Ref #167]

BALB/c mice - femoral artery ligation 7 days post-surgery

Elampretide

-decreased limb necrosis

-Improved mitochondrial respiration

-increase limb perfusion recovery and capillary density

Wilson et al. [Ref #168]

Mice – acute ischemia (1hr) and 7-14 days post-ischemia

MitoSNO

-increased muscle contractile function

-decreased muscle denervation

Clinical Studies in Human PAD Patients

Brevetti et al. [Ref #170-172]

IC patients (n=8)

IC patients (n=10)

IC patients (n=30)

L-carnitine

-did not change ABI

-improved walking distance

Brevetti et al. [Ref # 173]

IC Patients (n=485)

Propionyl-L-carnitine

-improved walking distance in severe claudicants but not mild

McDermott et al. [Ref #175]

IC Patients (n=44)

Epicatechin

-improved 6-min walk performance

-increase cytochrome c oxidase activity

-improved muscle perfusion and capillary density

Park et al. [Ref #176]

IC Patients (n=22)

mitoQ

-improved endothelial function

-improved walking distance and time

-delayed onset of claudication

  • Since this review deals with PAD and skeletal muscle ischemia, I would suggest to the authors to include at least a few sentences or a paragraph to the hypoxia-inducible factor (HIF). HIF has an important role in regulating mitochondria and angiogenesis. The review would gain if HIF is put into the context.

We appreciate this comment from the reviewer regarding HIF signaling in PAD. We agree this signaling pathway is likely to be an important part of the limb tissue biology in PAD.  However, we feel that a discussion of HIF would warrant a substantial expansion to orient the readers to HIF signaling which cannot be done in a brief manner. We feel that this level of addition and discussion of angiogenesis would distract readers from our goal of evaluating the role of muscle mitochondria in PAD pathobiology. We feel that keeping the review focused on mitochondria in muscle will provide a better “take-home” message for readers.

Reviewer 3 Report

I was honored to review the manuscript entitled “Skeletal muscle mitochondrial dysfunction and oxidative stress in peripheral arterial disease: A unifying mechanism and therapeutic target” submitted to Antioxidants. The study presents high quality and deals with important clinical issue, such type of study is needed.  I have only few small remarks that authors should address properly.

I recommend to accept the manuscript after minor revision.

There are only some points to correct:

 - please provide the list of abbreviations

- introduction and discussion section need improvement – please provide information on how your results will translate into clinical practice

- in discussion section please provide study strong points  and study limitation section

- please correct typos

 - It would be also useful to illustrate some of the explained mechanisms as in general the manuscript is quite long and therefore a little bit difficult to follow.

I recommend to accept the manuscript after minor revision.

Author Response

I was honored to review the manuscript entitled “Skeletal muscle mitochondrial dysfunction and oxidative stress in peripheral arterial disease: A unifying mechanism and therapeutic target” submitted to Antioxidants. The study presents high quality and deals with important clinical issue, such type of study is needed.  I have only few small remarks that authors should address properly.

I recommend to accept the manuscript after minor revision.

There are only some points to correct:

 1) please provide the list of abbreviations

We have followed the journal guidelines of defining abbreviations in parentheses at first use. Unless the journal recommends an abbreviations list, we aim to adhere to their stated instructions.

2) introduction and discussion section need improvement – please provide information on how your results will translate into clinical practice

3) in discussion section please provide study strong points  and study limitation section

For both comments #2 and 3 we are respectfully not sure exactly how to respond. This was a review manuscript discussing the literature and future directions regarding the role of muscle mitochondrial pathology in peripheral artery disease.  Based on this objective, we are not sure an expanded introduction or limitations section are appropriate.

- please correct typos

We have corrected all typographical errors in the revised manuscript.

 - It would be also useful to illustrate some of the explained mechanisms as in general the manuscript is quite long and therefore a little bit difficult to follow.

As mentioned above, we are not entirely sure how to address this comment from the reviewer. The goal of this review was to discuss the state of knowledge about muscle and mitochondrial pathology and its role in peripheral arterial disease. We focused these efforts on first providing background in PAD diagnosis, pathophysiology, and treatment.  Next, we thoroughly discuss the literature on muscle pathology, mitochondrial dysfunction, and oxidative stress.  Finally we conclude by highlighting preclinical, risk factor, and future treatment directions that can target the mitochondrial pathology in PAD.